# Long-Term Trends of Air Pollution at National Atmospheric Observatory Košetice (ACTRIS, EMEP, GAW)

**Milan Váňa [1,*], Adéla Holubová Smejkalová [1,2], Jaroslava Svobodová [1] and Pavel Machálek [3]**

[1] Czech Hydrometeorological Institute, Košetice Observatory, 394 22 Košetice, Czech Republic; adela.holubova@chmi.cz (A.H.S.); jaroslava.svobodova@chmi.cz (J.S.)

[2] Institute for Environmental Studies, Faculty of Science, Charles University, 128 01 Prague, Czech Republic

[3] Czech Hydrometeorological Institute, 143 06 Prague, Czech Republic; pavel.machalek@chmi.cz

*   Correspondence: milan.vana@chmi.cz; Tel.: +420-725-895-574

**Abstract:** The National Atmospheric Observatory Košetice operated by the Czech Hydrometeorological Institute was established in 1988 as a station specializing in air quality monitoring at the background scale. The observatory is located in the free area outside of the settlement and represents the Czech Republic in various international projects. The objective of the present study is to detect the long-term trends of air quality at the background scale of the Czech Republic. The statistical method used for trend analysis is based on the nonparametric Mann–Kendall test. Generally, the results show that the fundamental drop in emission of basic air pollutants was reflected in the significant decrease in pollution levels. A most significant drop was detected for sulphur. No trend was found for $NO_2$ in 1990–2012, but a visibly decreasing tendency was registered in the last 7 years. A slightly decreasing trend was registered for $O_3$ in the whole period, but a slightly increasing tendency was found after 2006. More importantly, the number of episodes exceeding the target value for human health dropped significantly. The reduction of volatile organic compounds (VOCs) emissions was reflected in a statistically significant decrease of concentrations. Only isoprene, which is of natural origin, displays an inverse trend. Concentrations of elemental carbon (EC) and organic carbon (OC) dropped since 2010, but only for EC is the trend statistically significant.

**Keywords:** long-term trends; background scale; air quality; Czech Republic

## 1. Introduction

The danger caused by large-scale, global and regional pollution started to be recognised in the 1960s. Such pollution might end up resulting in irreversible changes in both terrestrial and ocean ecosystems and global climate change. The research and monitoring efforts required to detect the changes in the atmosphere at global and regional scales must be based on broad-ranging international cooperation. It was, first of all, international institutions (World Meteorological Organization, United Nations Economic Commission for Europe ECE, United Nations Environment Programme) that initiated, in the 1960s and 1970s, the first international monitoring programmes [1].

To support the above-mentioned programmes, Czech Hydrometeorological Institute (CHMI) established the National Atmospheric Observatory Košetice (NAOK), specialized in monitoring and research of air quality at the background scale of Czech Republic.

After the political changes in 1989, the air quality control and protection became one of the most important political priorities in the Czech Republic. Immense funds were invested in emission reductions (mainly from large power plants) in the Czech Republic during the 1990s, resulting in a



marked improvement in the air quality, the levels of which in some regions had previously ranked among the worst in the world. Nevertheless, the growing industry and traffic after 2000 have caused the air quality in the Czech Republic to begin to deteriorate again. Irresponsible conduct of individuals who use low-quality fuels or even municipal waste in household heating systems, emitting hazardous chemicals to the air, is a contributing factor that cannot be neglected. Fine dust is the most serious problem at the moment. The Ministry of the Environment developed a National Emission Reduction Programme of the Czech Republic in 2007, and it has been approved by the government. The document comprises several key measures to contribute to an improvement in the current state of the environment and environmental and health protection.

The objective of the study is to detect the long-term trends of air quality at the background scale of the Czech Republic. Thirty-year data series is sufficient for detection of long-term trends of air quality. The study is based on the data generated within the National Air Pollution Monitoring Network, stored in Air Quality Information System and annually published e.g., [2]. Generally, the development of air quality in the last three decades was affected by various circumstances: the essential political changes in Central and Eastern Europe in the end of the 1980s brought a substantial decrease in emissions in the Czech Republic and more widely in the Central European region thanks to international conventions and also economic and political development. The meteorological conditions for long-range transport in Europe were changed as well, and the question of global climate change assumed importance. Measurement techniques showed significant improvement, as did our knowledge concerning the behaviour of air pollutants in the atmosphere. All these aspects influenced the long-term trends at the background scale of the Czech Republic very significantly.

## 2. Materials and Methods

### 2.1. Site Description and Overall Context

NAOK is located in the agricultural countryside outside of settlements in the southern part of the Czech Republic, district of Pelhřimov (49°35′ N, 15°05′, 534 E m asl, Figure 1). More detailed description of physical-geographical conditions is available in [3]. The operation of NAOK started in 1988, but some basic air quality measurements were implemented since the middle of the 1980s in the vicinity of the observatory. The main task of NAOK throughout its history was to detect the long-term trends of air quality at the background scale of the Czech Republic and Central Europe and to represent the Czech Republic in the long-term programmes of air quality monitoring and research GAW/WMO (Global Atmosphere Watch), EMEP (Co-operative Programme for Monitoring and Evaluation of Long-range Transmission of Air Pollutants in Europe) and ICP–IM (International Co-operative Programme on Integrated Monitoring).

After 2004, when Czech Republic joined the EU, NAOK, thanks to its excellent location and long-term homogeneous data series, has been participating in several EU projects. The first was EUSAAR (European Supersites for Atmospheric Aerosol Research), focused on the research of atmospheric aerosols. The essential importance for the advancement of NAOK in the last decade brought participation in ACTRIS RI (Aerosol, Clouds and Trace gases Research Infrastructure). NAOK is a core of Large Research Infrastructure (LRI) ACTRIS Czech Republic (ACTRIS-CZ), a unique platform for the long-term background air quality monitoring and research closely related to climate, environmental and health issues qualified as societal challenges. ACTRIS-CZ represents a national node of the existing European ACTRIS Research Infrastructure (RI) established with the support of the EU 7th Framework Programme INFRA-2010-1-1.1.16 (EU FP7) and ACTRIS-2 project of EU Horizon 2020 (H2020-INFRAIA-2014-2015: Integrating and Opening Existing National and Regional Research Infrastructures of European Interest). In December 2015, ACTRIS was adopted on the ESFRI roadmap 2016 for Research Infrastructures. LRI ACTRIS-CZ RI is based on the long-term collaboration of 4 research partners: Czech Hydrometeorological Institute (CHMI), The Institute of Chemical Process Fundamentals of the CAS (ICPF), Global Change Research Institute of the CAS (GCRI) and Masaryk University (MU) at the research facility of NAOK.

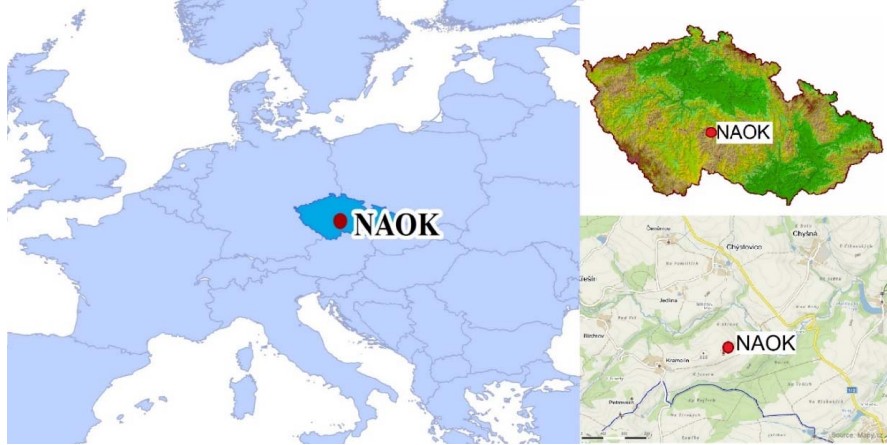

**Figure 1.** Location of National Atmospheric Observatory Košetice (NAOK) in the European, Czech and local context.

### 2.2. Measurement Methods

NAOK is a part of the National Air Pollution Monitoring Network (operated by CHMI). All measurements are carried out according to the quality-controlled procedures. This network operates online (automatic analyzers) and offline (samplers) instruments. Filter analyses are done mainly in the central laboratory of immissions in Prague. The data validation procedure is in line with the EC directive 2008/50/EC. An overview of methods of measured components used in this study are listed in Table 1. A detailed description of each method is available in [4].

**Table 1.** Used sampling methods—year of start, type of measurement, method of determination.

| Component | Start of Measurement | Type of Measurement | Method |
| :---: | :---: | :---: | :---: |
| CO | 1996 | online | IR corel. absorption spectrometry |
| EC/OC | 2009 | offline | heat decomposition_FID |
| $NO_X$ | 1994 | online | chemiluminescence |
| $O_3$ | 1992 | online | UV-absorption |
| $PM_{10}$ | 1996 | online | radiometry |
| $PM_{2.5}$ | 2004 | offline | gravimetry |
| $SO_2$ | 1992 | online | UV-fluorescence |
| $SO_4$ | 1988 | offline | ion chromatography |
| VOCs | 1995 | offline | gas chromatography |
| $\Sigma NH_4$ * | 2002 | offline | spectrophotometry |
| $\Sigma NO_3$ * | 2002 | offline | ion chromatography |

* The sum consists of gaseous and particulate matter.

### 2.3. Statistical Evaluation

The statistical method used for the evaluation of long-term trends is based on the nonparametric Mann–Kendall test for the trend and the nonparametric Sen's method for the magnitude of the trend. Mann–Kendall test is used since missing values are allowed and the data need not conform to any particular distribution. Sen's method is not greatly affected by gross data errors or outliers, and it can also be computed when data are missing. Sen's estimator is closely related to the Mann–Kendall test. Mann–Kendall test is recommended and plentifully used for long-term trend evaluation in air quality because it enables one to evaluate non-complete data series and does not require specific distribution of measured values. On the other hand, it is not possible to use it for assessment of short-term variability and annual variation. Due to its properties, the Mann–Kendall test was used to analyze long-term trends of air pollution measurements data in a harmonized way for EMEP Assessment report [5].

The presence of trend is evaluated using Z value. A positive and negative value of Z indicates an upward and downward trend, respectively. The statistics Z has a normal distribution.

The existence and significance of trend is tested by using four different $\alpha$ levels of significance. The different $\alpha$ levels used are $\alpha = 0.1$, $\alpha = 0.05$, $\alpha = 0.01$ and $\alpha = 0.001$.

Trend statistics it is given as a result of the significance level of that trend, marked by

+    if there is a trend at the $\alpha = 0.1$ level,

*    if there is a trend at the $\alpha = 0.05$ level,

**    if there is a trend at the $\alpha = 0.01$ level and

***    if there is a trend at $\alpha = 0.001$ level.

This means that when the mark is "***", the trend is very significant, and when the mark is "+", the significance of the trend is fairly poor, only 10%. If the mark is missing, then there is no trend at significance level $\alpha = 0.1$ [6].

## 3. Results

Political and economic changes after the fall of the iron curtain brought a general drop in industrial production and later significant changes in the structure of the industry. These changes were reflected in the reduction of sulphur emissions in the Czech Republic by almost 90% in the period of 1990–2000 (Figure 2). The results of long-term monitoring show that the emission decrease was reflected in reduced pollution levels. Sulphur dioxide concentrations in the atmosphere declined nearly by the same order of magnitude as the emissions (Figure 3c). The steep drop of $SO_2$ concentrations was more pronounced in the 1990s. The frequency of episodes with extremely high concentrations decreased rapidly (Figure 3e). In the new millennium, the mean annual concentrations dropped below 5 $\mu g \cdot m^{-3}$, but a slightly decreasing trend was found also in the period 2001–2019 (Figure 3f). In the EMEP domain, $SO_2$ emission reductions started in the 1980s–1990s; therefore, changes in concentrations will have occurred earlier than 1990. However, concentrations have continued to decrease continuously during the period under review. The timing of concentration decreases varies between countries according to national implementation of emission reduction strategies, but on average, the decrease was larger in the early 1990s and levelled off since then [5].

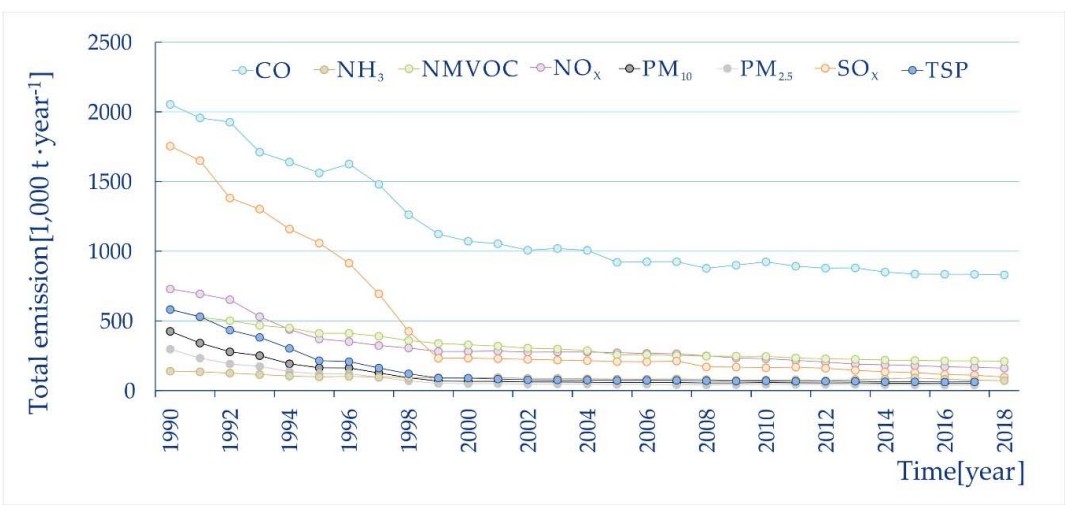

**Figure 2.** Total emissions of basic pollutants in the Czech Republic in the period 1990–2018. Results are based on emission inventories outcomes, regularly evaluated and published in air pollution reports in the Czech republic (e.g., [2]).

In the period of 1994–2012, no trend of nitrogen oxides concentration was found, in spite of the fact that the nitrogen emissions declined by 54% during the period under review (Figure 3b). In the period of 1990–2012, the emissions dropped by 72% and in the EU by 55% [6]. Mean annual concentrations varied around 10 $\mu g \cdot m^{-3}$. These results were in very good correspondence with the trends at the

background level in the neighbouring countries (Austria, Germany) [5]. The reasons are uncertain. One of the explanations could be the significant changes in the structure of nitrogen emissions. In the last 7 years, a visibly decreasing tendency of $NO_2$ concentrations was found and the mean annual concentrations dropped continuously to 4 $\mu g \cdot m^{-3}$. The evaluation of the data from the EMEP network shows that for the period of 1990–2001, the fraction of sites where significant negative trends were observed was high (58%), but it slowed down after 2002.

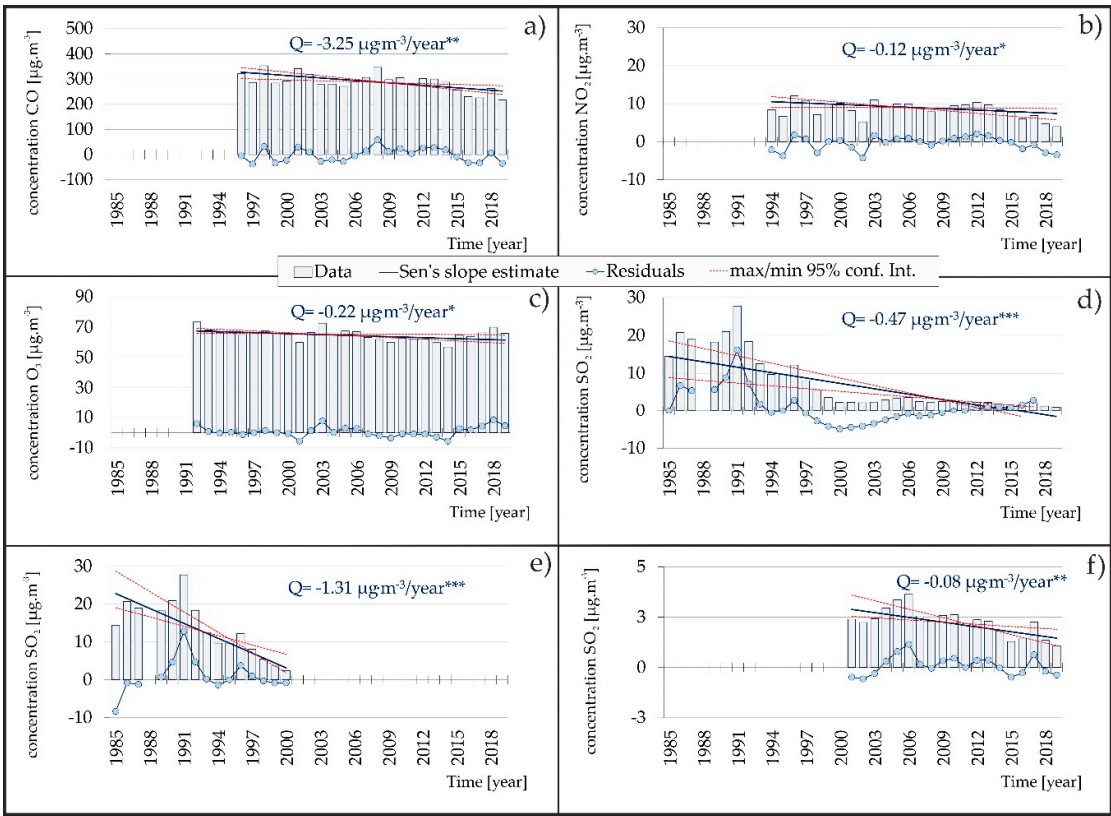

**Figure 3.** Results of Mann–Kendal test for gaseous pollutants; (**a**) CO, (**b**) $NO_2$, (**c**) $O_3$, (**d**) $SO_2$ 1985–2018, (**e**) $SO_2$ 1985–2000, (**f**) $SO_2$ 2001–2018.

NO concentration at the background scale is quite low, and mean annual concentrations varied around 1 $\mu g \cdot m^{-3}$. The long-term trend describes similar patterns for $NO_2$: no trend in the period 1994–2012 and decreasing tendency after 2012.

A slightly decreasing trend was found in mean annual concentrations of tropospheric ozone in the whole period and also in the first part of the period under review (Figure 3c). On the contrary, a slightly increasing tendency was found after 2006. It is caused probably by increasing temperature during the last two decades. A warm period displays similar patterns as the whole year. On the contrary, no trend was found in the cold period [7]. More importantly, the number of episodes exceeding the target value for human health dropped significantly during the period (Figure 4), and interannual variations can be explained by meteorological conditions. The target value of tropospheric ozone for the protection of human health is exceeded when the eight-hour running mean is higher than 120 $\mu g \cdot m^{-3}$ 25 times on average for 3 years. Visibly higher values were recorded in the years with extreme summer temperatures over long periods and well-established heat waves over continental Europe (2003, 2015, 2017).

A statistically significant trend was found for carbon dioxide. Mean annual concentrations decreased continuously during the whole period (Figure 3a).

Most non-methane volatile organic compounds (VOCs) follow an annual course that reflects their emission levels, i.e., with maximums in winter and minimums in summer. Isoprene is an exception.

In general, the reduction of VOCs emissions in the last two decades was reflected in a decrease of concentrations at the regional scale of the Czech Republic [8]. A statistically significant downward trend was found for almost all of measured VOCs, and only the ethane trend was less significant (Table 2). The trend of isoprene concentrations is controlled first of all by natural conditions and shows different patterns from other VOCs. We detected a statistically very significant upward trend of isoprene concentration in the period under review. Favourable conditions for isoprene emissions are in hot summer periods. An increasing tendency was visible especially in the last decade. This is in good correlation with increasing mean annual temperature in the current period of changing climate conditions (hot summers, long periods with high temperatures). It follows from the current report on VOC measurements in the context of EMEP [9] that the VOC concentrations continuously decrease on a regional scale and thus reflect the decreasing trend in emissions. The concentration level at NAO Košetice is comparable with those at the German, Swiss and French stations. The Czech station has long been characterised by lower annual average ethane concentrations. For most VOCs, the concentrations measured in the winter are usually similar to those at German stations, while the values at NAOK are slightly lower in the summer.

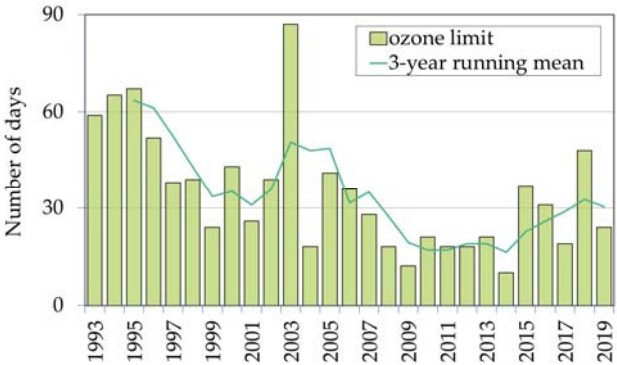

**Figure 4.** Number of days with target limit for surface ozone exceedances (1993–2019).

**Table 2.** Trend significance of measured VOCs.

| Time Series | First Year | Last Year | n | Test Z | Signific. | Q |
|---|---|---|---|---|---|---|
| Acethylene | 1993 | 2018 | 26 | −5.16 | *** | −46.91 |
| Benzene | 1993 | 2018 | 26 | −3.92 | *** | −8.11 |
| Butenes | 1993 | 2018 | 25 | −2.06 | * | −2.93 |
| Cyklohexane | 1996 | 2018 | 23 | −2.85 | ** | −1.13 |
| Cyklopentane | 1995 | 2018 | 22 | −3.13 | ** | −0.69 |
| Ethane | 1993 | 2018 | 26 | −1.94 | + | −12.71 |
| Ethene | 1993 | 2018 | 26 | −4.89 | *** | −28.23 |
| Ethylbenzene | 1993 | 2018 | 25 | −0.86 | | -0.68 |
| i-Butane | 1993 | 2018 | 26 | −5.73 | *** | −6.56 |
| i-Octane | 1997 | 2018 | 21 | 1.60 | | 0.24 |
| i-Pentane | 1993 | 2018 | 25 | −5.63 | *** | −9.92 |
| Isopren | 1993 | 2018 | 26 | 3.66 | *** | 1.95 |
| m,p-Xylene | 1993 | 2018 | 25 | −0.30 | | −0.18 |
| Metylcyklopentane | 1996 | 2018 | 22 | −3.70 | *** | −1.26 |
| n-Butane | 1993 | 2018 | 26 | −4.94 | *** | −11.28 |
| n-Hexane | 1993 | 2018 | 25 | −3.95 | *** | −3.15 |
| n-Octane | 1997 | 2018 | 22 | 0.57 | | 0.22 |
| Nonane | 1996 | 2018 | 22 | −0.73 | | −0.21 |
| n-Pentane | 1993 | 2018 | 25 | −5.11 | *** | −5.88 |
| o-Xylene | 1993 | 2018 | 25 | 0.21 | | 0.22 |
| Pentenes | 1996 | 2018 | 23 | −2.03 | * | −0.67 |
| Propane | 1993 | 2018 | 26 | −4.74 | *** | −14.20 |
| Propene | 1993 | 2018 | 26 | −4.79 | *** | −4.73 |
| Toluene | 1993 | 2018 | 26 | −4.63 | *** | −8.28 |

"***"—the trend is very significant; "+"—the significance of the trend is fairly poor, only 10%. If the mark is missing, then there is no trend at significance level $\alpha$ = 0.1.

The measurement of aerosol particles covers periods of different duration. The longest records are available for sulphate in aerosol and $PM_{10}$. Table 3 shows that the concentrations were changed significantly during the period under review. Changes in concentration levels reflect the development of emission (Figure 2), which is in line with both national and international environmental measures.

Outcomes of sulphur and $PM_{10}$ show the highest level of trend significance (Figure 5c,d). Sulphur concentration continuously decreased during the whole period. On the other hand, the evaluation of $PM_{10}$ data shows that the mean annual concentrations in the period of 2001–2006 reached a similar level as in 1996 (29.8 $\mu g \cdot m^{-3}$) (Figure 5c). The same patterns were observed across the Czech Republic at different types of stations. After 2001, the drop of emission was slower compared to the previous period. The increase of $PM_{10}$ concentrations was probably influenced by meteorological and dispersion conditions [10]. A higher level of trend significance is observed for $PM_{2.5}$ concentrations. After 2005, when a level over 18 $\mu g \cdot m^{-3}$ was recorded, the linear decreasing trend is observed. These outcomes are analogous for $PM_{10}$.

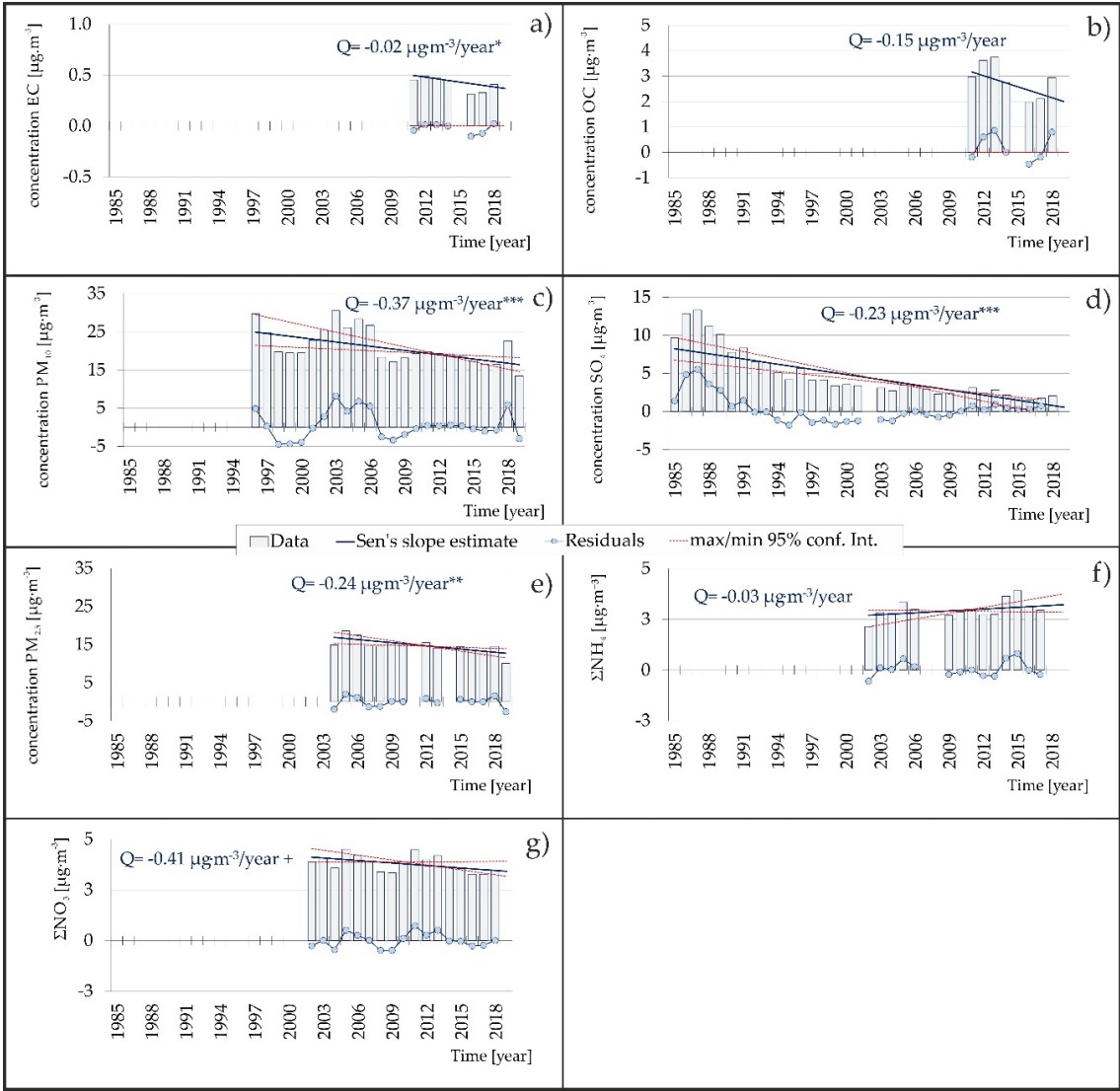

**Figure 5.** Results of Mann–Kendal test for aerosol particles; (**a**) EC, (**b**) OC, (**c**) $PM_{10}$, (**d**) $SO_4$, (**e**) $PM_{2.5}$, (**f**) $\sum NH_4$, (**g**) $\sum NO_3$.

An insignificant trend is visible for the sums of ammonium and nitrates (Figure 5f,g) that are measured from 2002. This is in line with the fact that the emission development is more or less at the same level (Figure 2). No visible annual variation was found (Table 3). Concentrations of elemental

(EC) and organic carbon (OC) dropped from 2010 (Table 3), but only for EC is the trend statistically significant (Figure 5a,b).

**Table 3.** Changes in annual aerosol concentrations at the beginning and the end of the evaluated period.

| Component | EC | OC | $PM_{10}$ | $SO_4$ | $PM_{2.5}$ | $\Sigma NH_4$ | $\Sigma NO_3$ |
|---|---|---|---|---|---|---|---|
| concentration [$\mu g \cdot m^{-3}$] | | | | | | | |
| Beginning of period | 0.5 | 3.3 | 29.8 | 9.6 | 14.9 | 2.1 | 3.9 |
| End of period | 0.4 | 2.9 | 13.4 | 2.1 | 10.1 | 2.9 | 3.5 |

## 4. Summary

Generally, the results show that the fundamental drop in emission of basic air pollutants in the Czech Republic and widely in the Central European region in the period under review was reflected in the significant decrease of air pollution levels at the background scale of the Czech Republic. A statistically very significant drop in mean annual concentrations of sulphur dioxide was detected in the period of 1990–2000. After 2000, the mean annual concentrations dropped below 5 $\mu g \cdot m^{-3}$, but a slightly decreasing trend was found also in the period of 2001–2019.

No trend was found by the evaluation of nitrogen dioxide in the atmosphere in the period of 1990–2012, in spite of the fact that the nitrogen emissions declined by half during the period under review. In the last 7 years, a visibly decreasing tendency of $NO_2$ concentrations was registered, and the mean annual concentrations dropped continuously to 4 $\mu g \cdot m^{-3}$.

A slightly decreasing trend was found in mean annual concentrations of tropospheric ozone in the whole period and also in the first part of the period under review. On the contrary, a slightly increasing tendency was found after 2006. It is caused probably by increasing temperature during the last two decades. More importantly, the number of episodes with the target value for human health exceedances dropped significantly during the period.

The reduction of VOCs emissions in Central Europe was reflected in a statistically significant decrease of concentrations at the regional level of the Czech Republic. Only isoprene, which is of natural origin, displays an inverse trend.

Sulphur concentration in aerosol continuously decreased during the whole period. The evaluation of $PM_{10}$ data shows that the mean annual concentrations in the period of 2001–2006 reached a similar level as in 1996. The higher level of trend significance is observed for $PM_{2.5}$ concentrations, but the general outcomes are analogous as for $PM_{10}$. Concentrations of EC and OC dropped from 2010, but only for EC is the trend statistically significant.

**Author Contributions:** Conceptualization, M.V.; methodology, A.H.S.; validation, M.V., A.H.S., J.S.; formal analysis, M.V., A.H.S.; investigation, M.V., A.H.S.; resources, M.V.; data curation, J.S., P.M.; writing—original draft preparation, M.V.; writing—review and editing, M.V.; visualization, A.H.S.; supervision, M.V.; project administration, M.V.; funding acquisition, M.V. All authors have read and agreed to the published version of the manuscript.

**Funding:** This research was funded by the project for support of national research infrastructure ACTRIS—participation of the Czech Republic (ACTRIS-CZ - LM2015037)—Ministry of Education, Youth and Sports of the Czech Republic.

**Acknowledgments:** The research leading to these results has received funding from the project for support of national research infrastructure ACTRIS—participation of the Czech Republic (ACTRIS-CZ - LM2015037)—Ministry of Education, Youth and Sports of the Czech Republic.

**Conflicts of Interest:** The authors declare no conflict of interest.

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
