# Peer review of "Long-Term Trends of Air Pollution at National Atmospheric Observatory Košetice (ACTRIS, EMEP, GAW)"

_atmosphere, doi:10.3390/atmos11050537_

Round 1

Reviewer 1 Report

Comments and suggestions

1. Reformulate the sentence (line 99-100):

Suggestion: Due to its properties, the Mann-Kendal test was used to analyse long-term trends of air pollution measurement data in a harmonised way for the EMEP Assessment report in [5].

2. Figure 2. The figure is too busy. Legend has the same colours (red) for 1990 and 2000. The trend is obvious but the graph is made without care. I suggest that all years except 1990, 2000 and 2005 were presented in one colour (for instance, light gray) while depicted years 1990, 2000 and 2010 in red, blu and green thus indicating by colour and magnitude changes in emissions over the period 1090-2018. This would make a graph look better and more readable.

3. In the lines, 126-134 discussion of trends in NO2 concentrations is not properly explained with the reduction of NOx emissions. Indeed, a strong decline in emissions (78% in CR for the whole period) is not valid for the period 1990-2012. A decline in these years was around 50% not only in CR, but in Europe as well. Moreover, the decline of NOx emissions was much slower or even leveled up after the first rapid decline. This is the cause of the features of the trend in NO2 concentrations, not the structural redistribution between different emission categories (industry vs. traffic). This structural change exists in all countries but does not explain the trend. Trend explains the fact that NOx emissions did not drop drastically as it was the case with SO2 emissions. Please, revise the reasoning accordingly.

4. Ozone: line 140-145. and Figure 4.: The graph is not easy to understand in the context of trends. Three years are especially worth mentioning: 2003, 2015 and 2018. Namely, these three years were years with extreme summer temperatures over long periods and well-established heath waves over continental Europe. Data show it. It is worth mentioning the influence of meteorological conditions, especially on ozone exceedances. Also, during the last decade, some years had very expressed minima, which also coincides with unusually wet and colder summers. These variations make trends variable, but these interannual variations can be explained by meteorological conditions.

5. Biogenic emissions of isoprene and its significant upward trend should be commented in connection with changing climate conditions (hot summers, long periods with high temperatures). This should be substantiated by a few relevant references.

Author Response

Most of suggested remarks and comments were respected in the updated text. The figure was changed according to the recommendation. Regarding point 3. (nitrogen emissions): The drop of emission in the Czech Republic 1990-2012 was 72% (regular inventory of the Department of Emissions and Sources of the Czech Hydrometeorological Institute). The CZ emission data was prepared specially for this article. The drop of European Emissions in the same period was 55% (source cited in the new text). The significant changes in the structure of nitrogen emissions was one of the explanations of “no trend” of NO2 during our discussions at least at the national level. But it is for sure not the only one reason. I changed the formulation in the new version.

Reviewer 2 Report

Comments on atmosphere-773527

Because this article has many flaws, I would like to request for tremendous revisions, especially for the presentation quality. Please see below comments.

Major comments:

Figure 2: I do not follow the meaning of green colors in this figure, and it is not understandable because each bar is so small. Please revise this figure for the readability. In addition, what is the data source for this emission estimations?

Figure 3 and its relevant discussion: Did (d), (e), and (f) show SO2 concentrations? This seems to be related to the discussion in P4, L114-122\; however, there is no clear sentences to state them. First of all, Figure 3d should be discussed before the separate discussion on 3e and 3f.

Figure 4 and its related discussion: The discussion in P5, L142-144 is just the speculation, and it seems not good to just refer. How about to add the temperature at this site and show its relation?

Minor comments:

P3, Table 1: What is the meaning of summation for NH4 and NO3?

P3, L104: Typo of “α = 0.05? Please confirm the expression as * defined in the following sentence.

P5, L144-145: What is the target value?

Technical comments:

P4, L110-112: Do not use bold type here.

Throughout this manuscript: Please unify to use the abbreviation for air pollutants.

Author Response

The figure 2. And Fog. 3 was changed according to the recommendation. Summation for NH4 and NO3 is explained by the note under the table. Description of target values of ozone exceedances was included. Minor and technical comments respected in the updated version. Regarding nitrogen emissions and concentrations: The drop of emission in the Czech Republic 1990-2012 was 72% (regular inventory of the Department of Emissions and Sources of the Czech Hydrometeorological Institute). The CZ emission data was prepared specially for this article. The drop of European Emissions in the same period was 55% (source cited in the new text). The significant changes in the structure of nitrogen emissions was one of the explanations of “no trend” of NO2 during our discussions at least at the national level. It is not just a speculation, but on the other hand it is for sure not the only one reason. I changed the formulation in the new version.

Reviewer 3 Report

Long-term trends of air pollution at National Atmospheric Observatory Košetice (ACTRIS, EMEP,GAW)

The manuscript is simple report on long-term measurement of air quality, conducted by the CHI at the background locality Kosetice.

The report presents interesting data but must be corrected for several following flaws and omissions:

Introduction: too short, have to discuss air quality at the background locality with respect to evolution of air quality in the CR at least; short intro about long-term trends of the pollutants at background localities at the neighboring states would help to formulate working hypothesis, and justify selection of appropriate tool for the data evaluation. Working hypothesis is missing.

L.50-51. Explain how the improvement of our knowledge on behavior of air pollutants influenced the pollutant long-term trends.

Experimental

Detailed map of the site would be helpful.

Statistic evaluation:

Integration time and number of the data in each dataset for specific year/pollutant have to be present.

Without this information it is difficult to evaluate quality of consequential data analysis.

L122. Fi2. 2. Probably cut-and-paste form another publication; what means green colour in the graph?

L126. No trend in nitrogen oxides was found = statement before data analysis?

Fig 3. Bad quality, difficult to read, why there are halfwidth bars for 1996 or 2019? Difficult to compare when different time scale was selected for each graph.

L137. NO trends: discussion of the values close to the detection limits is missing

L140 Define what means “slightly decreasing trend…” also further in the text

Extreme ozone value in 2003? Any explanation? Is it exemption? Is the reason too hot and dry summer? It needs explanation how this was evaluated.

Table 3. Are there annual values for the start and end for specific components? But the same information can be seen in the Figure 3.

Figure 3. Bad quality, difficult to read, different time scales, again why there are halfwidth bars for 1996 or 2019.

L 189 Distinctively statistically significant drop…..Requires English correction

Since the term “mean annual concentrations” appears in the text frequently, annual averages were evaluated in this manuscript, while it was not erroneously stated in the appropriate section. But, Mann-Kendall test is sensitive to seasonality, i.e. periodic changes within the year, hidden in the annual value. It will be more useful and convenient to present and evaluate trends for seasonally segregated averages.

To conclude, the presented manuscript does not fulfill the criteria for scientific publication and I recommend to rewrite it thoroughly and resend.

Author Response

Some of the comments and remarks were respected. The paragraph regarding air quality development and policy in the CR was included. The comparison with the trends in the EMEP domain was included into the main text. The detailed map of the site was included.

Integration time and number of the data in each dataset for specific year/pollutant have to be present.   Without this information it is difficult to evaluate quality of consequential data analysis. The start of each measurement was already indicated in the original version in Tab. 1. No change at this stage.

Fi2. 2. Probably cut-and-paste form another publication; what means green colour in the graph? The figure was changed. The data was prepared specially for this article. We would like to stress that there was no cut-and-paste form. It is just the wrong meaning of the reviewer.

All contested figures were updated.

More detailed explanation of ozone trends was included into the updated version. Regarding nitrogen emissions and concentrations: The drop of emission in the Czech Republic 1990-2012 was 72% (regular inventory of the Department of Emissions and Sources of the Czech Hydrometeorological Institute). The CZ emission data was prepared specially for this article. The drop of European Emissions in the same period was 55% (source cited in the new text). The significant changes in the structure of nitrogen emissions was one of the explanations of “no trend” of NO2 during our discussions at least at the national level. It is not just a speculation, but on the other hand it is for sure not the only one reason. I changed the formulation in the new version.

Generally, this review differs from two others very much. It seems to me as non-objective and unfriendly.

Round 2

Reviewer 2 Report

The manuscript partly improved; however, I guess that there are still some problems especially on the presentation quality. I am concerning the following two points.

  1. Analysis on emissions: For example, in L149-150, It is stated that "In the period 1990-2012, not trend of nitrogen oxides concentrations was found, in spite of the fact that the nitrogen emissions declined by 78% during the period under review (Figure 3b)", but is it nitrogen oxides showed no trend? It seems to be the slight decline. In addition, the observation started from 1994, and hence this sentence is ambiguous. The analysis on emissions which consistent to the analyzed period of observation is required for clear discussion.
  2. Figure 3 and Figure 5: I appreciate the revision for time with same period, and I have further request. Fig. 3f, Fig. 5a, d, f, g showed so small bar graphs, and it is ambiguous to figure out the changes in concentrations. It will be better to fit y-axis for each figure based on the maximum concentration of each specie. 

Author Response

  1. Analysis on emissions: For example, in L149-150, It is stated that "In the period 1990-2012, not trend of nitrogen oxides concentrations was found, in spite of the fact that the nitrogen emissions declined by 78% during the period under review (Figure 3b)", but is it nitrogen oxides showed no trend? It seems to be the slight decline. In addition, the observation started from 1994, and hence this sentence is ambiguous. The analysis on emissions which consistent to the analyzed period of observation is required for clear discussion.

It is true: The monitoring of NO2 started in 1994. The opening sentence of the paragraph was change in following manner: In the period 1994–2012, no trend of nitrogen oxides concentrations was found, in spite of the fact that the nitrogen emissions declined by 54% during the period under review (Figure 3b). This correction does not have any influence on the whole paragraph regarding NO2

  1. Figure 3 and Figure 5: I appreciate the revision for time with same period, and I have further request. Fig. 3f, Fig. 5a, d, f, g showed so small bar graphs, and it is ambiguous to figure out the changes in concentrations. It will be better to fit y-axis for each figure based on the maximum concentration of each specie. 

Figs. 3 and 5. Were changed according to the recommendation.

Round 3

Reviewer 2 Report

I appreciate the authors to address my concerns.